# An effort-based social feedback paradigm reveals aversion to popularity in socially anxious participants and increased motivation in adolescents

**Dienke J. Bos**[1,2]*, **Emily D. Barnes**[1], **Benjamin M. Silver**[1], **Eliana L. Ajodan**[1], **Elysha Clark-Whitney**[1], **Matthew A. Scult**[1], **Jonathan D. Power**[1], **Rebecca M. Jones**[1]

**1** Department of Psychiatry, Sackler Institute for Developmental Psychobiology, Weill Cornell Medicine, New York, NY, United States of America, **2** Department of Psychiatry, UMC Utrecht Brain Center, University Medical Center Utrecht, Utrecht, The Netherlands

\* d.j.bos-2@umcutrecht.nl

## Abstract

We created a novel social feedback paradigm to study how motivation for potential social links is influenced in adolescents and adults. 88 participants (42F/46M) created online posts and then expended physical effort to show their posts to other users, who varied in number of followers and probability of positive feedback. We focused on two populations of particular interest from a social feedback perspective: adolescents relative to young adults (13–17 vs 18–24 years of age), and participants with social anxiety symptoms. Individuals with higher self-reported symptoms of social anxiety did not follow the typical pattern of increased effort to obtain social feedback from high status peers. Adolescents were more willing to exert physical effort on the task than young adults. Overall, participants were more likely to exert physical effort for high social status users and for users likely to yield positive feedback, and men were more likely to exert effort than women, findings that parallel prior results in effort-based tasks with financial rather than social rewards. Together the findings suggest social motivation is malleable, driven by factors of social status and the likelihood of a positive social outcome, and that age, sex, and social anxiety significantly impact patterns of socially motivated decision-making.

## Introduction

Social feedback is fundamental to human interactions and decision-making, but modes and scales of social feedback are evolving rapidly with the proliferation of social media platforms. Users of social media can find themselves rapidly exposed to large amounts of feedback by virtue of their links in social networks. For example, a comment on a news item may be highlighted by an account with many followers, triggering a torrent of support or criticism. Or, an aspiring artist may awake to find that an influential account highlighted their work, which was then accessed or purchased by tens of thousands of people overnight. Such events

**Data Availability Statement:** The minimal anonymized dataset is available on OSF (DOI: 10.17605/OSF.IO/73P9T).

**Funding:** We would like to thank the Mortimer D. Sackler, MD family for generous funding support. The authors received no other specific funding for this work.

**Competing interests:** The authors have declared that no competing interests exist.

can have substantial effects on users' emotions, careers, and lives. Such feedback is mediated by links in a large network, and the choices users make about linking to other accounts in the network therefore have practical importance in terms of potential downstream benefits or harms the user might experience. All other things being equal, accounts with many followers can instigate rapid and extensive feedback compared to accounts with fewer followers, and accounts likely to give positive feedback are inherently desirable.

To examine how potential social feedback influences the links individuals choose in social networks, we designed a paradigm analogous to the effort expenditure for rewards task (EEfRT). The EEfRT measures how much motoric effort humans are willing to expend to obtain monetary rewards [1, 2], and shows that participants reliably exert more effort for larger or more probable monetary rewards. Notably, males choose the higher effort task more often than females [1]. One potential explanation may be that males have increased sensitivity to (monetary) rewards compared to females [3–5], but whether the EEfRT task is gender-biased is an open question [1]. In our study, participants created social media posts and had to expend physical effort to show their posts to accounts with variable numbers of followers and variable probabilities of positive feedback. It took more physical effort to show posts to accounts with many followers or high probability of positive feedback, and the task was titrated so that subjects exhibited fatigue over the experiment, indicating they had to weigh whether to (continue to) give effort for rewards.

We focused on behavioral effects in two populations of particular interest from a social feedback perspective: effects in adolescents relative to young adults, and effects in participants with social anxiety symptoms. Adolescence is a transformative period characterized by profound social development [6–8]. During adolescence the importance of peer relationships strongly increases [9], yet with simultaneously growing self-consciousness [10, 11] adolescents find themselves highly sensitive to peer approval and rejection [12–15]. As such, adolescents may be more motivated to obtain social evaluative information [13, 16]. Previous work using physical exertion as a measure of motivation has shown that both adolescents and adults expend increasing effort to obtain increasing monetary rewards [17]. However, given that adolescents are also more sensitive to online peer acceptance and rejection [10, 18], adolescent users of social media might show greater motivation, i.e. willingness to expend effort, for positive social feedback, especially from those with high social status, than adults.

Yet, social change in adolescence may coincide with the emergence of social anxiety [19–21]. Individuals with social anxiety are averse to strangers and scrutiny and fear potential embarrassment or humiliation. Socially anxious individuals seem to fear social evaluation in general, both negatively and positively [22, 23]. Socially anxious youth also experience less positive affect after positive events or social interactions, and they may avoid potential positive interactions altogether to prevent disappointment or embarrassment [24, 25]. While adolescence is a phase of hypersensitivity to social evaluation in general, responsivity to social feedback seems to change as a function of age and severity of social anxiety [26, 27]. For instance, compared to older socially anxious youth (13–18 years), young socially anxious adolescents (8–13 years) were more responsive to unpredictable negative and predictable positive social evaluations, as indicated by more extreme pre-task ratings on peer likeability and greater activity in dorsal anterior cingulate cortex and insula during anticipation of social evaluation, suggesting increased salience of anticipated social interactions in young adolescents [28]. Users of social media with social anxiety may avoid linking to accounts with large numbers of followers, thereby limiting potential harm (and benefit), and they may be especially likely to seek ties likely to give positive feedback. Conversely, given that social milieus now increasingly assume an online form [29, 30], social anxiety may be attenuated in the anonymity of online interactions [31]. Whereas classically socially anxious individuals were described as people likely to

avoid parties or public speaking, socially anxious individuals may show no differences in online social media interactions to those without anxiety. Because online social interactions are now an integral part of personal and professional life, it is important to understand what motivates or inhibits socially anxious youth and young adults so that mitigation strategies can be targeted (e.g., to prevent avoidable professional limitation).

Our primary aim in this study was thus to develop an ecologically valid effort-based paradigm for assessing choices to pursue positive peer feedback. We expected, in parallel with prior results in effort-based monetary reward paradigms [1], to observe increased effort for accounts with more followers, increased effort for higher probability of positive feedback, and increased effort in males compared to females. We also expected disparities in how adolescents and young adults expended effort given disparities in the financial reward literature [17]. Additionally, we anticipated potential disparities between anxious and non-anxious individuals. Demonstration of such social decision-making effects lays the foundation for studies of commonalities and distinctions among reward modalities in motivating behavior, the individual biases that favor or disfavor particular kinds of rewards, and the neurobiological underpinnings of such biases.

## Materials and methods

### Participants

111 13–24 year olds (54 females) were recruited through the Sackler Institute for Developmental Psychobiology in Manhattan, New York. Prior to participation, participants (or caregivers, if participant was under 18 years old) confirmed no prior history of psychiatric disorders and no current medications relating to a psychiatric illness, via telephone. Three participants were excluded due to task-related technical difficulties, five participants were excluded for being unable to complete the task motorically, two participants were excluded due to failure to follow task instructions, and 13 participants did not believe the cover story and were excluded from all analyses. See Table 1 for the demographics of the 88 participants included in analyses.

**Table 1. Sample characteristics.**

|  | Adolescents (13–17 years) | Young Adults (18–24 years) |
|---|---|---|
| **N** | 40 | 48 |
| **Males/Females** | 23/17 | 23/25 |
| **Age (mean, std)** | 14.6 (1.3) | 21.0 (1.7) |
| **Peterson Puberty Scale**[*] | Males Females | n/a |
|  | 3.4 (0.7) 3.5 (0.4) |  |
| **Ethnicity** |  |  |
| White (%) | 48 | 30 |
| Hispanic (%) | 32 | 21 |
| Asian (%) | 0 | 29 |
| African American (%) | 10 | 10 |
| More than one race (%) | 10 | 10 |
| **Verbal IQ** | 109.45 (17.9) | 113.8 (17.6) |
| **Nonverbal IQ** | 100.2 (23.4) | 105.5 (13.0) |
| **LSAS Total Score** | 40.3 (26.5) | 35.2 (20.2) |

[*] There was no significant difference in pubertal development between males and females (p = 0.618). Abbreviations: LSAS = Liebowitz Social Anxiety Scale Self-Report, IQ = Intelligence Quotient

## Procedures

Participants completed cognitive testing, questionnaires, and the social effort task in one visit. Participants ages 18 years and older provided informed consent and participants 13–17 year provided assent and their caregivers provided informed consent. Participants were compensated $40. The Weill Cornell Medicine Institutional Review Board approved the protocol.

Young adult participants completed the Liebowitz Social Anxiety Scale Self-Report (LSAS-SR; [32, 33]) and adolescent participants completed the LSAS for Children and Adolescents Self-Report (LSAS-CA-SR; [34]) to assess social anxiety. The child and adult versions of the LSAS both yield scores with 24 rated items on a 0–3 Likert scale. The Peterson Puberty Scale was administered to caregivers of 13–17 year olds to assess pubertal development. A research assistant administered the Differential Ability Scales-II (DAS-II; [35]) verbal and nonverbal reasoning subtests to adolescents ages 13–15 years and the Wechsler Adult Intelligence Scale-IV (WAIS-IV; [36]) verbal comprehension and perceptual reasoning subtests to participants ages 16 years and older. There were no significant differences between adolescents and young adults in Verbal IQ, Nonverbal IQ, or LSAS Total Score ($p$'s $> 0.206$).

Participants also completed three pre-task questions to determine the subjective value of receiving social media likes in general, the importance of the quantity of likes received, and the importance of the social status of the people liking the post. The three questions were: *How much does it matter to you when someone likes your post on social media*? *When you post something on social media, how much does it matter to you how many likes you get*? *When someone likes your post on social media, how much does it matter to you how many followers that person has*?. The three questions were rated on a 7-point Likert-scale (1- extremely unimportant to 7-extremely important). These questions were intended to capture differences in values that may influence how participants made decisions during the social effort task [37].

## Cover story and instructions

Participants were given a digital camera and told to take five photographs in the waiting room. Participants were informed that these photos would be seen by people on a website called M-Turk, so they should feel confident about the photos that they took. Participants were unfamiliar with M-Turk and were told that M-Turk is a website that is often used in psychology research and that approximately 3,000 people had signed up on M-Turk to help with the present study. After participants completed taking the five photos, they were told that their photos would be uploaded to M-Turk, and that upon uploading, all participating M-Turk users would get a notification that the study was about to begin, and that whoever was available should log on. The purpose of this cover story was to enhance the believability that the participants would be interacting with peers who had evaluated them in some manner.

Task instructions were reviewed with the participant on a laptop screen prior to beginning the task. Participants were shown a screen with the M-Turk logo and reminded that M-Turk is often used in psychology research because it allows researchers to get access to many people quickly. Participants were told that a series of M-Turk users would view one of their photos chosen at random, and that the M-Turk user would provide binary feedback (like or not like) on the photo.

For each trial, participants had to choose one of two M-Turk users that they wanted to view their photo (Fig 1). The participant was told how many followers the M-Turk users had on the social media website Instagram, and the probability that the M-Turk user would provide positive feedback (a like) on the participant's photo, based upon the total number of photos that the M-Turk user liked for all prior participants in the study.

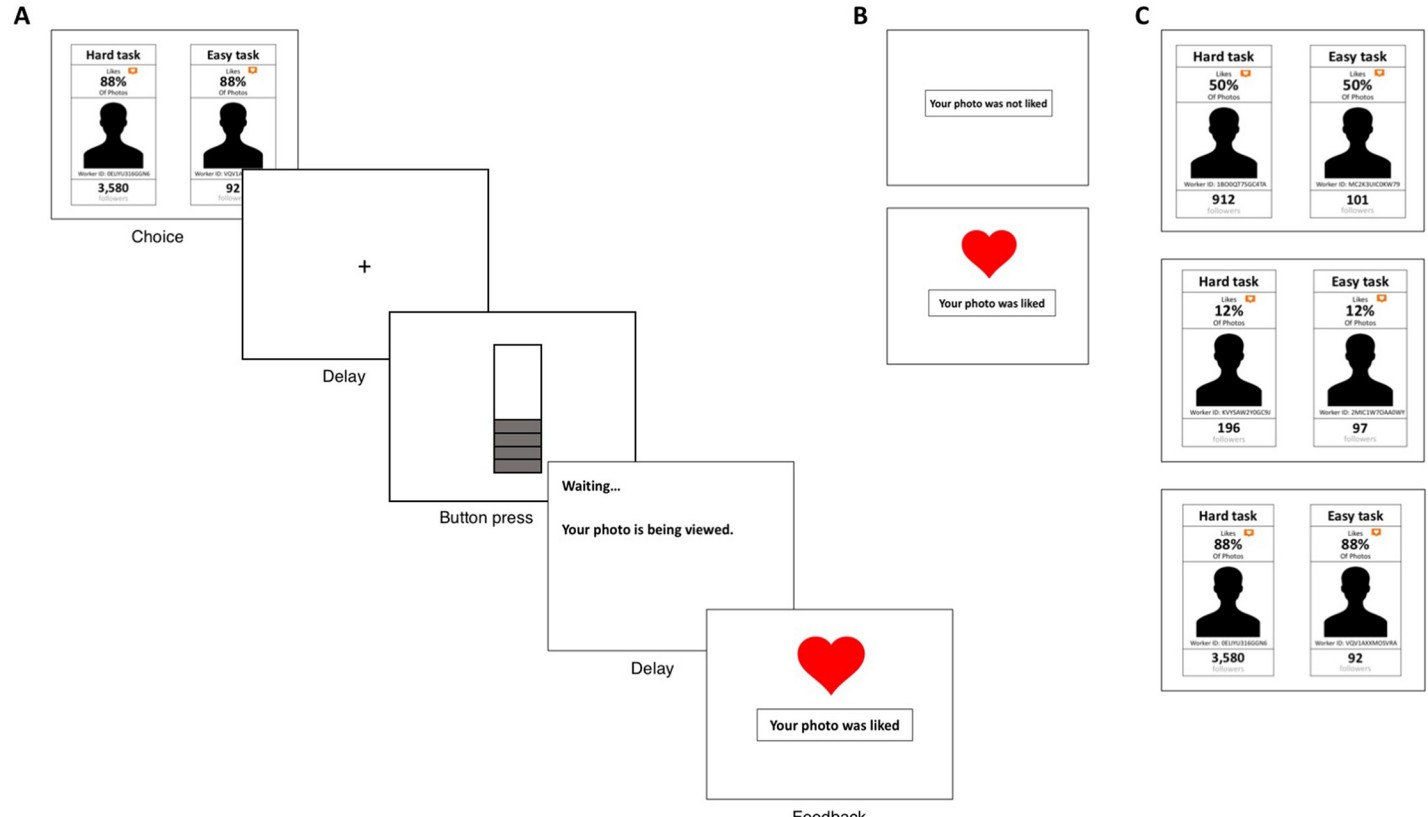

**Fig 1. Experimental paradigm.** A) Sequence of a single trial. Participant is presented with a choice between an easy task and a hard task. After choosing and after a 1000 ms delay, participants completed the button-pressing task. Upon completion of the task, participants waited three seconds for feedback. B) Negative feedback (top) and positive feedback (bottom). C) Three examples of task choices; 50% probability of a like (top), 12% probability of a like (middle), and 88% probability of a like (bottom).

Participants were told that in order to have their photo viewed, they would have to complete either an easy button-pressing task or a hard button-pressing task (differentiated by the finger used to complete the task and the number of button-presses required; see *Task* section for more detail) on the computer keyboard. Each M-Turk user was associated with either the easy task or the hard task. Participants were told that if they successfully completed the button-pressing task that they chose, the associated M-Turk user would view their photo in real time and would then have three seconds to provide feedback.

Participants were told to try their best to complete all trials and that it was against the rules to fail on purpose or to use the wrong finger. Participants were told that the game would take 20 minutes to complete, regardless of the choices that they made. There were four practice trials before the task began.

After completing the task, participants responded to three questions rated on a 7-point Likert scale that assessed the subjective importance of receiving a like during the task and how positive or negative participants felt when they received a like and when they did not. The three questions were: *During the task, how much did it matter to you when someone liked your photo? How did you feel when someone liked your photo? How did you feel when someone did not like your photo?*.

To assess whether participants believed the premise of the task, each participant was asked if they thought that each person viewing their photos during the game was a real person.

Participants were asked the following question: *Did you think all the people liking your photos were real people*?. Participants were then debriefed.

## Social effort task

The social effort task was programmed in MATLAB using the Psychophysics Toolbox version 2.0 and presented on a laptop. In order to bridge our study of social reward to prior studies of financial reward, we created a paradigm with straightforward parallels to effort-based monetary feedback paradigms. One of the best-studied tasks for examining financially-motivated decisions is the effort expenditure for rewards task (EEfRT), which was designed to assess how much motoric effort humans are willing to expend to obtain monetary rewards [1, 2]. In EEfRT studies, participants must make a choice of whether to exert higher effort (pressing a button with non-dominant pinky finger many times) or lower effort (pressing a button with dominant index finger fewer times) for rewards of varying probability (12%, 50%, 88%) and magnitude ($1.24-$4.30). Our social feedback task had a parallel structure such that the probability of receiving money was replaced with the probability of receiving positive feedback in the form of a like from an M-Turk user, while monetary reward was replaced with the M-Turk user's social status, represented by their number of Instagram followers.

There were three levels of probability of positive feedback: low (12% probability), medium (50% probability), and high (88% probability). Analogous to the classic EEfRT, the easy task was always associated with an M-Turk user who had a very low number of Instagram followers (90–110), in order to make sure that the easy task was the least appealing compared to the hard task choices. The hard task M-Turk users varied between 124–4,120 followers, divided into three social status levels for analyses: low (124–220 followers), medium (660–1,020 followers), and high (3,400–4,120 followers). There are thus 3 levels of **probability** of positive feedback, and 3 levels of **social status** of the account accessed by the hard task, which form the basis of statistical contrasts of behavior below. After four practice trials, there were 54 trials, with 18 trials for each probability level. There were 18 trials of the low social status level, 19 trials of the medium social status level, and 17 trials of the high social status level.

To complete the hard task, participants were required to press a keyboard key 98 times within 21 seconds, using their non-dominant pinky finger. To complete the easy task, participants were required to press a keyboard key 30 times within seven seconds, using their dominant index finger. Each key press raised the level of a bar on the screen, and the participant was successful on the trial if they filled the bar to the top within the allotted time. Upon successful completion of the task, there was a three-second delay before positive or negative feedback was given to the participant.

## Preprocessing

Participant choice, whether the trial was completed, whether positive or negative feedback was received, social status level, and probability level were saved from Psychtoolbox on each trial and imported into MATLAB (version R2017b). The proportion of times that the hard task was chosen was calculated for the entire task. The proportion of times the hard task was chosen was also calculated for each probability level (12%, 50%, and 88%), each social status level (low, middle, and high), and for each of the nine combinations of the two factors (3 probability levels by 3 social status levels). These values were also calculated for the early (first 18 trials), middle (middle 18 trials), and late (last 18 trials) trials of the task.

Data were exported from MATLAB into R (version 3.5.1) for statistical analysis. When participants did not make a choice within seven seconds, the trial was excluded from analyses as it was considered a missed trial. Of the 88 participants, 64 had 0 missed trials, 23 had 1–3 missed

trials, and 1 had 4 missed trials. For each of the probability levels and social status levels, as well as for the nine combinations of levels, the proportion of hard-task choices was calculated as the raw number of times the hard task was chosen in each condition over the total number of trials in that condition.

### Statistical analyses

In the first analyses, age was tested by binning subjects into groups. To determine differences in task performance between age groups, we chose to define adolescents as 13–17 years old and young adults as 18–24 years old to be consistent with prior literature [37)], although see [38]. We conducted a 3x3 repeated measures ANOVA in R version 3.2.1 (R Core Team, 2015) with the proportion of hard task choices as the dependent variable and two within-subjects factors of social status level (low, medium, high) and probability level (12%, 50%, 88%), with age (adolescent, young adult) and sex (male, female) as between-subjects factors. Significant main effects and interactions ($p < 0.05$) were further interrogated using least-square means. The analysis was repeated with age as a continuous variable.

Second, for the social anxiety analyses, scores from the LSAS-SR and the LSAS-CA-SR were divided into two groups with a clinical cutoff of 60 [39, 40]. Using this cutoff, 75 participants (32 adolescents (30 female) and 43 young adults (22 female)) fell into the non-elevated range (total scores ranging from 0–59) and 13 participants (8 adolescents (all female) and 5 young adults (3 female)) fell into the elevated range (total scores ranging from 60–103). Social anxiety level (non-elevated, elevated) was then added to 3x3 repeated measures ANOVA as a between-subjects factor. Significant interactions ($p < 0.05$) were followed up using least-square means. To confirm our findings, we performed two follow-up tests: First, we divided our sample with a different clinical cutoff of 47. A cut-off of 60 has been suggested to be the best cut-off for identifying generalized Social Anxiety Disorder (SAD), yet maximal sensitivity for generalized SAD was obtained using a cut-off of 47 [39, 40]. Given that this is a non-clinical sample of participants we chose to run all analyses using both cut-offs. Using a cutoff of 47, 59 participants (25 adolescents (24 female) and 34 young adults (17 female)) fell into the non-elevated range and 29 participants (15 adolescents (14 female) and 14 young adults (8 female)) fell into the elevated range. We repeated all analyses with this second clinical cutoff. Finally, social anxiety level was added to the analysis as a continuous variable. There were no meaningful differences between the results of the analyses using the clinical LSAS cutoff of 60, using a LSAS cutoff of 47, or including LSAS scores as a continuous variable (see S1–S7 Tables).

To ensure that groups valued the task equally, responses on the three pre-task and three post-task questions were compared by independent sample t-tests. There were no statistical differences between adolescents and young adults in responses to the six questions ($p$'s > 0.12), no significant differences between males and females ($p$'s > 0.06), and no significant differences between the anxiety-defined groups ($p$'s > 0.24).

Secondary analyses focused on whether there was fatigue over time on the task. To assess fatigue, a repeated measures ANOVA was conducted examining the proportion of hard-task choices by time (early, middle, late) on the task. To determine if fatigue was impacted by age, sex, or social anxiety, between-subjects factors of age, sex, and social anxiety level were added separately to the repeated measures ANOVA.

## Results

In our social feedback task we found main effects of social status and probability. Participants chose the hard task more when there was a higher probability of receiving positive feedback ($F$ (2, 688) = 20.9, $p < .001$) and when the M-Turk User had a higher social status ($F$ (2, 688) =

48.7, $p < .001$). There was also a main effect of age group ($F (1, 84) = 4.0$, $p = .048$) and sex ($F (1,84) = 10.0$, $p = .002$), where adolescents and males respectively chose the hard task more often. As there were no three- or four-way interactions (all p's > .349) these were consequently dropped from the design. There was an interaction between social status and probability ($F (4, 688) = 2.7$, $p = 0.032$) (Fig 2), as participants chose the hard task more often when it was highly likely that they would receive positive feedback from high social status peers. There was also an interaction of sex and social status such that males chose the hard task more often for high social status peers ($F (2, 688) = 11.4$, $p < .001$). All other two-way interactions were not significant (p's > .084) See S1 and S2 Tables for full statistics and post-hoc comparisons.

Secondary analyses including age as a continuous variable did not meaningfully change the results. The interaction between social status and probability ($F (4, 688) = 2.7$, $p = 0.032$) and between sex and social status remained ($F (2, 688) = 11.3$, $p < .001$). See S3 Table for full statistics.

The analysis including social anxiety level showed that participants with higher symptoms of social anxiety gave the same effort overall as non-anxious participants (i.e, there was no effect of social anxiety on proportion of choosing the hard-task, $p = .821$). There was no interaction of social anxiety and probability ($p = .713$), yet there was an interaction between social anxiety and social status ($F (2, 684) = 5.3$, $p = .005$) (Fig 3). See S4 and S5 Tables for full statistics and post-hoc comparisons. Post-hoc comparisons showed that there were no significant group differences between those who did or did not have elevated social anxiety. However, within groups, adolescents and young adults without social anxiety symptoms chose the hard task more often with increasing social status, whereas those with elevated social anxiety symptoms did not differentiate by social status in their hard task choices. Analyses were repeated at a different LSAS cutoff of 47 (a more inclusive definition of social anxiety) and with LSAS scores as a continuous variable, which yielded the same effects (see S6 and S7 Tables).

It was important that our participants weighed the worth of their effort in order to properly detect motivation via willingness to perform button presses; in this sense it was desirable that participants exhibit fatigue. Indeed, participants chose the hard task most often in the early trials, less in the middle trials, and least often in the late trials ($F (2,174) = 21.8$, $p < .001$) (S1 Fig, S8 Table). There was no interaction of task section with sex, age, or social anxiety levels (p's > .090).

## Discussion

The present study developed an effort-based social feedback task meant to mimic interactions in social media networks, and is notable for several reasons. First, for establishing that participants exert more physical effort for high-status social links (e.g., friends, followers) and for links likely to yield positive feedback, and that men are more likely to exert effort for a social reward than women. These results establish a bridge between the financial reward and social reward literature. Second, for establishing that sex, age, and symptoms of social anxiety affect the choices participants make about which kinds of links to pursue. The findings together suggest social motivation is malleable, driven by factors of social status and the likelihood of a positive social outcome, and that age, sex, and social anxiety significantly impact patterns of socially motivated decision-making.

Overall, high social status increased the likelihood that participants would choose the hard task. These findings are consistent with financial incentive tasks in which participants give more effort for larger magnitude rewards [41, 42], with research indicating that adolescents and young adults are drawn to photographs on Instagram by popularity [43, 44], and that visual attention is greater for high- versus low-status peers [45–47]. The probability of a

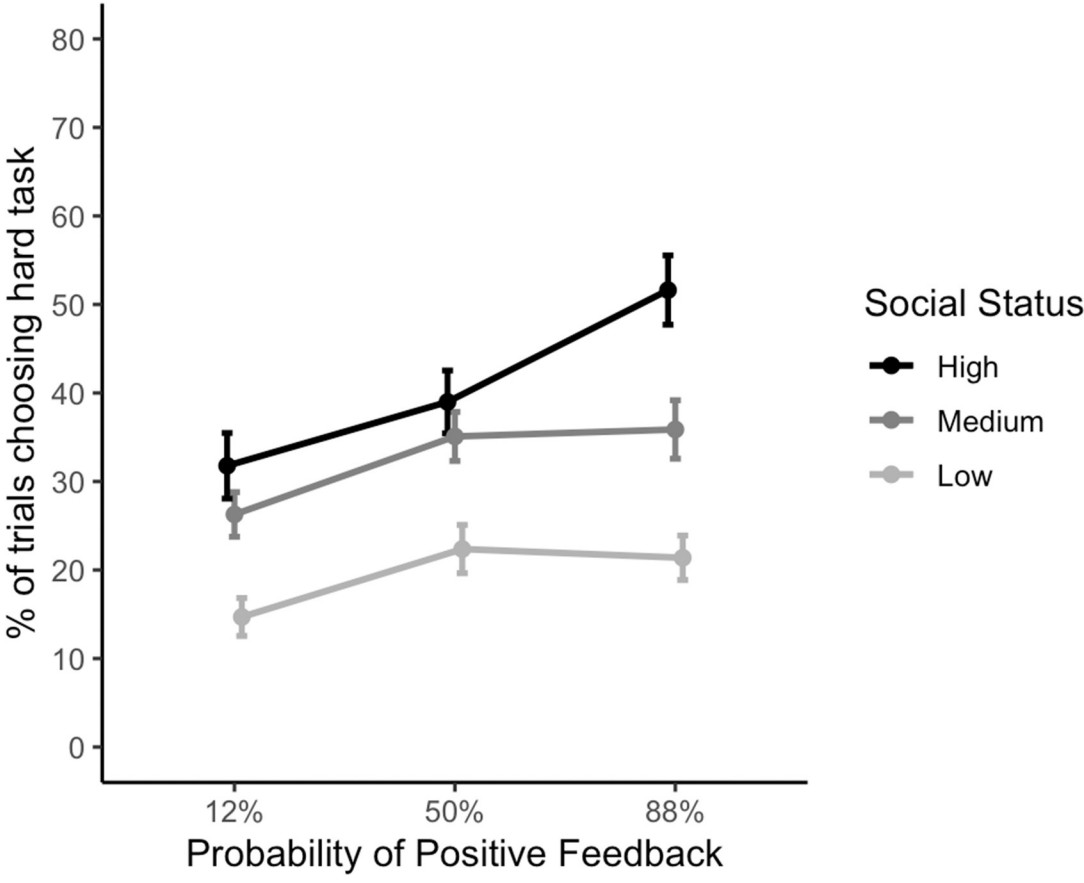

**Fig 2. Number of hard task choices.** Hard-task choices by probability and social status trial conditions.

positive reward also increased the decision to choose the hard task, consistent with earlier work using the original monetary version of the EEfRT [2, 48–50]. As such, our findings suggest that we developed a successful social adaptation of the original monetary EEfRT. Previous work using e.g. social incentive delay paradigms [51–54] or social feedback paradigms [28, 55–58] have shown effects of adolescence or social anxiety on the processing of social evaluative information. Our social effort paradigm extends this work by capturing the motivation or willingness to expend effort to obtain social feedback. Yet importantly, how we value different types of social or non-social rewards, and consequently, how much effort we are willing to expend is influenced by individual characteristics such as age, gender, or social interaction- or anxiety problems [51–53, 59, 60]. Below we will highlight several notable modulations of status pursuit in our data, all of which deserve further study.

Only adolescents displayed an exception to a monotonic increase of effort with probability of positive feedback, displaying the highest effort when the probability of positive feedback was 50%. The 50% trials have the least assured outcome, and, are in that sense, the riskiest trials. As we found no self-reported differences in how adolescents value social media feedback compared to adults, it seems unlikely that the increased effort by adolescents was simply driven by a difference in social valuation, and we instead interpret the finding as potentially reflecting a difference in social motivation, consistent with findings from behavioral and neuroimaging studies. Interacting with peers is more salient and activating for adolescents [13, 61] and adolescents are more willing to take risks in social contexts [62–64]. Adolescence is a phase that is

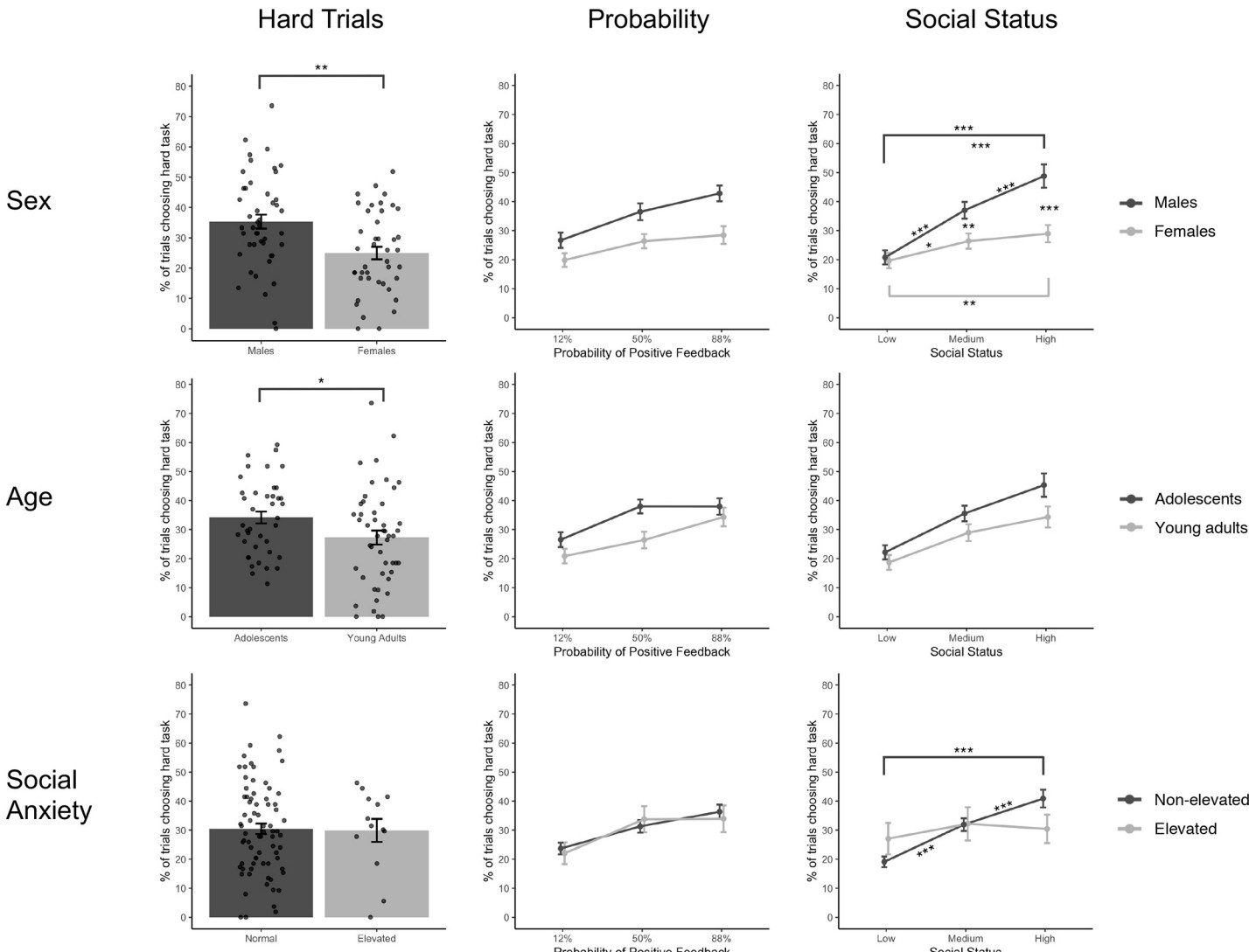

**Fig 3. Hard task choices by sex, age, and social anxiety.** Hard task choices (mean ± SEM) by sex, age, and social anxiety levels across all trials, by probability conditions, and by social status conditions. Significant main effects and interactions noted along with all statistics in S1, S2, S4 and S5 Tables. Asterisks denote significance at $^*$p < .05, $^{**}$p < .01 and $^{***}$p < .001.

in general associated with heightened activity of reward and socioemotional processing circuitry [8], acceptance from peers activates dopamine rich frontostriatal circuitry [65], and adolescent risk-taking in social contexts has been associated with increased engagement of reward, limbic, and salience circuitry [66–69]. The neural mechanism for the financial effort tasks, similar to our paradigm, relies on dopamine circuitry [41, 70] and circuitry for valuation and salience [71], which all experience protracted development throughout adolescence into early adulthood. Second, males were substantially more likely than females to choose high-status associated with the hard task (50% vs 25% at the high-status level, respectively), though females still demonstrated a differential effect of status (comparing low-to-high). The sex difference is compatible with studies finding that men tend to be more competitive in physical effort-based tasks [72], yet rewards in these studies were monetary. While the tendency for competitiveness in males may have been a factor driving the increased effort to obtain social

feedback, it has been shown that across age and between sexes there are behavioral and neuro-biological differences in how social or financial (or other) rewards are valued relative to each other [7, 51, 73, 74]. Future work that directly compares effort-based decision-making for social versus financial or other rewards can determine whether and how adolescence and/or sex influence motivation to obtain specific types of rewards [37].

A result of particular interest is that participants with higher levels of self-reported social anxiety displayed no modulation of hard task choice by status, in contrast to non-anxious par-ticipants, who scaled effort to status. This finding held using both conservative and liberal cri-teria to define anxious and non-anxious groups. The relative absence of differentiation in motivation, where socially anxious youth do not seem to follow the typical pattern of increased effort to obtain social feedback from high status peers, may be a safer or more secure social decision, and is consistent with prior work showing that individuals with higher anxiety symp-toms are more risk-averse [75] and show fear-avoidant decision-making behaviors [76]. Neu-roimaging studies have shown changes in striatal activity during positive and negative social feedback in socially anxious adolescents, which has been suggested to reflect increased affective arousal during social evaluation [28, 55]. Our findings may also reflect less context-appropriate social decision-making and an increase in rigidity in social scenarios, rather than flexibly adapting to varying feedback that one might receive during social interactions. Their altered decision-making pattern may also be maladaptive and perpetuate symptoms of loneliness and isolation, and thus may be an important target for interventions.

It is noteworthy that avoidance of popular accounts was seen in a "typical" population, for the present sample were all healthy controls, screened for any psychiatric disorders or medica-tions. Future work in individuals diagnosed with social anxiety disorder is needed to examine these behaviors in persons with more severe levels of social anxiety. It is also worth noting that there were no significant interactions between social anxiety and the probability of receiving positive social feedback or between social anxiety and sex in our data. The observation that in socially anxious individuals only the social status of the peer influenced effort, whereas proba-bility of positive social feedback did not, may suggest that also in social anxiety effort is mallea-ble, depending on context. Further, given the group sizes of individuals with elevated symptoms of social anxiety in the present study (conservative criterion size 13/89 subjects, 8 female, or liberal criterion size 29/89 subjects, 15 female), future work that explores the rela-tionship between social anxiety and sex on social motivation will be important, as there is gen-erally a higher preponderance of social anxiety in women.

Importantly, over all subjects, effort dropped over each third of the experiment. The magni-tude of the decrease in effort over a 20-minute task (~35% down to 25% on hard choices) sug-gests that the physical task is well-calibrated in the sense of truly causing subjects to fatigue and weigh the value of their exertions in making decisions. As there were no interactions of fatigue with age, anxiety, or sex, differential rates of fatigue do not explain any of our main findings.

Collectively, these findings suggest further behavioral studies, and set the stage for imaging studies that characterize the neurobiological underpinnings of such decisions. Future work that uses the social effort task during fMRI will provide information about what neural cir-cuitry is engaged during distinct social choices. Of particular interest will be the neural cir-cuitry of social effort-based decisions in individuals with social anxiety as compared to healthy controls. Such work will be critical for understanding a mechanism for why individuals with social anxiety show altered decision-making.

## Supporting information

**S1 Fig. Task fatigue.** Participants chose the hard task less often as the task progressed.
(DOCX)

**S2 Fig. Social anxiety and sensitivity to social status.** The difference in hard task choices between high and low social status trials by the LSAS total score. Higher difference scores indicate greater preference for high social status trials. Vertical lines indicate cutoff scores of 47 and 60 on the LSAS-SR.
(DOCX)

**S1 Table. Social effort task statistics.**
(DOCX)

**S2 Table. Social effort task statistics: Pairwise comparisons of significant main and interaction effects.**
(DOCX)

**S3 Table. Social effort task statistics with continuous age.**
(DOCX)

**S4 Table. Social effort task statistics with social anxiety.**
(DOCX)

**S5 Table. Social effort task statistics with social anxiety: Pairwise comparisons of significant main and interaction effects.**
(DOCX)

**S6 Table. Social effort task statistics with social anxiety at LSAS cut-off of 47.**
(DOCX)

**S7 Table. Social effort task statistics with continuous LSAS and age scores.**
(DOCX)

**S8 Table. Effects of fatigue.**
(DOCX)

## Acknowledgments

We would like to thank Faith Gunning for helpful discussions and Michael Treadway for sharing the monetary EEfRT task design.

## Author Contributions

**Conceptualization:** Emily D. Barnes, Benjamin M. Silver, Eliana L. Ajodan, Matthew A. Scult, Jonathan D. Power, Rebecca M. Jones.

**Data curation:** Emily D. Barnes, Benjamin M. Silver, Eliana L. Ajodan, Elysha Clark-Whitney, Rebecca M. Jones.

**Formal analysis:** Dienke J. Bos, Emily D. Barnes, Rebecca M. Jones.

**Funding acquisition:** Rebecca M. Jones.

**Investigation:** Emily D. Barnes, Eliana L. Ajodan, Rebecca M. Jones.

**Methodology:** Emily D. Barnes, Benjamin M. Silver, Eliana L. Ajodan, Elysha Clark-Whitney, Matthew A. Scult, Rebecca M. Jones.

**Project administration:** Emily D. Barnes, Rebecca M. Jones.

**Resources:** Rebecca M. Jones.

**Software:** Emily D. Barnes, Elysha Clark-Whitney, Rebecca M. Jones.

**Supervision:** Jonathan D. Power, Rebecca M. Jones.

**Validation:** Rebecca M. Jones.

**Visualization:** Elysha Clark-Whitney, Rebecca M. Jones.

**Writing – original draft:** Emily D. Barnes, Benjamin M. Silver, Jonathan D. Power, Rebecca M. Jones.

**Writing – review & editing:** Dienke J. Bos, Emily D. Barnes, Benjamin M. Silver, Eliana L. Ajodan, Elysha Clark-Whitney, Matthew A. Scult, Jonathan D. Power, Rebecca M. Jones.

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
