## [Decision Letter · Decision Letter 0]

7 Oct 2020

PONE-D-20-26279

An effort-based social feedback paradigm reveals aversion to popularity in socially anxious participants and increased motivation in adolescents

PLOS ONE

Dear Dr. Power,

Thank you for submitting your manuscript to PLOS ONE. After careful consideration, we feel that it has merit but does not fully meet PLOS ONE’s publication criteria as it currently stands. Therefore, we invite you to submit a revised version of the manuscript that addresses the points raised during the review process.

Two expert reviewers have evaluated your manuscript. Although both reviewers are enthusiastic about the results, they note some important concerns detailed below. For example, both reviewers recommend revising the introduction to clarify the theoretical frameworks and background that lead to the hypotheses. In addition, Reviewer 1 identified methodological issues relating to the analyses, including the use of categorical (rather than continuous) variables and also inconsistencies between the reported statistics and claims in the discussion and abstract. 

In addition to these issues, please ensure that  URLs/accession numbers/DOIs associated with data and materials are available or will be made available after acceptance so that we can ensure their inclusion before publication.

We look forward to receiving your revised manuscript.

Kind regards,

David V. Smith, Ph.D.

Academic Editor

PLOS ONE

Journal Requirements:

2.  Please change "Caucasian” to “White” or “of [Western] European descent” (as appropriate).

3. Please include in the manuscript additional information regarding the questionnaires used in the study and ensure that you have provided sufficient details that others could replicate the analyses. For instance, if you developed a questionnaire as part of this study and it is not under a copyright more restrictive than CC-BY, please include a copy, in both the original language and English, as Supporting Information. In addition, please provide additional details about the cognitive test assigned to the participants.

"We would like to thank the Mortimer D. Sackler, MD family for generous funding support."

Reviewers' comments:

Reviewer's Responses to Questions

**Comments to the Author**

1. Is the manuscript technically sound, and do the data support the conclusions?

Reviewer #1: Partly

Reviewer #2: Yes

2. Has the statistical analysis been performed appropriately and rigorously? 

Reviewer #1: Yes

Reviewer #2: Yes

3. Have the authors made all data underlying the findings in their manuscript fully available?

Reviewer #1: Yes

Reviewer #2: Yes

4. Is the manuscript presented in an intelligible fashion and written in standard English?

Reviewer #1: Yes

Reviewer #2: Yes

5. Review Comments to the Author

Reviewer #1: This is a well written manuscript describing a novel variant of the EEfRT task used to determine if the amount of effort expended varies by peer status and probability of positive feedback. Individual differences relating to sex, age, and social anxiety symptoms are explored. This new task is of sound design and makes an important contribution to the literature. However, use of categorical rather than continuous variables limit the power of some analyses, and interpretation of the social anxiety data need to be revised substantially to match the statistical results reported. This and other comments are described below.

Say more in the introduction about why it might be that males tend to choose the harder task more often in the monetary EEfRT task.

The introduction should do more to contextualize potential interactive effects of age and social anxiety for peer-based social cognition. For example, see Smith, Nelson, Kircanski et al., 2020.

Were there sex differences on the Peterson Puberty Scale? Was puberty controlled for in analyses that consider sex?

Were there scoring differences in the adult and child version of the LSAS? If so, how were those differences handled?

Please list the pre- and post-task questions verbatim. If a Likert scale was used, please describe the numeric and verbal anchors.

Was the deception question simply a yes or no question? If not, what was the operational definition of deception?

Please provide the logic for always assigning the easy task to the peer with a very low number of Instagram followers. Although this mirrors the classic EEfRT task, those unfamiliar with the task would benefit from further explanation of why it is set up this way.

Are results consistent when age is used as a continuous rather than categorical measure? Please report on whether they are consistent and describe inconsistencies in the supplement.

Likewise, are results consistent when the LSAS is treated as a continuous variable? Please indicate the N who fell above the clinical cutoff in each age group. No logic is provided for the use of two clinical cutoffs. Given there is a great deal of variability in anxiety symptoms in community samples, using the LSAS as a continuous variable would provide a more nuanced understanding of anxiety. It appears that one relation with LSAS as a continuous variable is provided in the supplement, but no mention of that occurs in the primary text.

Secondary analyses that include interactive effects of sex, age, and anxiety should be described in the methods section. Given the small number of cases that fell above the clinical cutoff, these secondary analyses should be performed using age and anxiety as continuous variables.

The significance indicators in Figure S3 do not always seem to map onto the lower order results described in the supplemental tables. For instance, social anxiety x probability data seem to be missing from the tables.

The authors characterize effects for social anxiety as follows: “Participants with higher symptoms of social anxiety avoided selecting high-status accounts and favored low-status accounts, relative to participants without symptoms of social anxiety, but differences only reached trend level in post-hoc comparisons (p’s > 0.084).” This does not fully capture the data described in the supplement – that there were no effects of social status among those with social anxiety. Only non-anxious individuals differentiated by social status. These results need to be better described in the text. The non-significant results currently highlighted in the text should be removed.

In the discussion the authors highlight the idea that men (change to males – these data average across children and adults) are more likely to exert more effort for a social reward than women (change to females). However, in the introduction the authors indicate the same relation is observed in the monetary domain. Thus, it is impossible to make a claim that this is specific to social experiences without a direct comparison between tasks. A direct comparison across social and monetary tasks is a critical next step that should be discussed. This is mentioned in the discussion, but could be highlighted further.

In the discussion, it is important to note that it appears that effort is malleable among non-anxious individuals. Anxious individuals seem relatively unaffected by contextual factors (at least based on the data reported about status in the supplemental materials – probability data were not reported – perhaps this is where this effect may have emerged).

This line in the discussion is quite misleading: “Second, participants with high anxiety scores, compared to non-anxious subjects, were more likely to select low-status and to avoid high- status choices, causing them overall to display no modulation of choice by status (in contrast to non-anxious participants, who scaled effort to status).” As described by the author’s own supplemental results, there were no significant between group differences. This type of language needs to be eliminated from the discussion and abstract. Effects were primarily driven by status-related differences within the non-anxious group. Also – please change the language from “normal” to “non-elevated”.

S

ocial Anxiety x Social Status

Elevated

Low vs. medium 12 0.91 0.379

Medium vs. high 12 0.27 0.790

Low vs. high 12 0.49 0.633

Normal

Low vs. medium 74 5.82 < 0.001 ***

Medium vs. high 74 3.58 < 0.001 ***

Low vs. high 74 6.46 < 0.001 ***

Low Normal vs. elevated 15 1.40 0.183

Med Normal vs. elevated 16 0.05 0.964

High Normal vs. elevated 23 1.80 0.085

“The increase in motivation for low social status peers may be a safer or more secure social decision, and is consistent with prior work showing that individuals with higher anxiety symptoms are more risk- averse (33) and show fear-avoidant decision-making behaviors (34).” It is not clear what increase in motivation the authors are referring to – there is no difference within the high anxiety group for any pair-wise comparison of status level, nor is there a difference between the low and high anxiety groups at any status level.

Reviewer #2: Thank you for the opportunity to review this manuscript entitled “An Effort-Based Social Feedback Paradigm Reveals Aversion to Popularity in Socially Anxious Participants and Increased Motivation in Adolescents” for consideration at PLOS ONE. As stated by the authors, this study created a novel social feedback program to assess motivational and effort-based ties to social links in adolescents and adults with and without high social anxiety symptoms.

This is an interesting topic, and in general I found this study to be a good contribution to the literature. Overall this manuscript is likely to be of interest to the broad audience at PLOS ONE, and has particular relevance for social media and technology-based approaches to future treatment. Having carefully considered this study, I provide (enumerated below) a few suggestions that may be of use to the authors in their efforts to further refine their manuscript. I provide one substantive comment regarding the introduction section below as well, that I highly suggest the authors consider:

1. Within the introduction, the authors identified potential differences in the literature that may contribute to adolescent versus adult social processing as well as some information regarding social anxiety and the impact that those symptoms may have on social processing. Although the authors present this information, it is extremely brief. I would strongly suggest that the authors significantly enhance their introduction section to set up a more solid rationale for the study. As it is currently written, they do not provide a compelling rationale that sets their study apart from others currently in circulation. Specifically, the authors should flesh out the theoretical underpinnings of adolescent versus adult social processing, as well as social feedback processing in individuals with social anxiety. Further, it would be beneficial for the authors to discuss other social feedback and decision-making paradigms that have been used in both adult and adolescent populations.

2. Somewhat related to my first comment above, the discussion section of this manuscript could be expanded significantly to reflect the novelty of the paradigm the authors have created, and how it could compare to currently existing paradigms. I suggest the author’s consider this point.

3. Within the discussion section, the authors jump to discussing neuroimaging studies using this paradigm. While I think this is more than appropriate, it does not directly link to the preceding sections. I suggest the authors thread this point into their discussion section more, and then elaborate on the novelty of this paradigm for neuroimaging studies.

6. PLOS authors have the option to publish the peer review history of their article (what does this mean?). If published, this will include your full peer review and any attached files.

Reviewer #1: No

Reviewer #2: No

---

## [Author Response · Author response to Decision Letter 0]

20 Jan 2021

Reviewer #1: This is a well written manuscript describing a novel variant of the EEfRT task used to determine if the amount of effort expended varies by peer status and probability of positive feedback. Individual differences relating to sex, age, and social anxiety symptoms are explored. This new task is of sound design and makes an important contribution to the literature. However, use of categorical rather than continuous variables limit the power of some analyses, and interpretation of the social anxiety data need to be revised substantially to match the statistical results reported. This and other comments are described below.

We thank the reviewer for their positive comments about the design of the task. We replicated our age and social anxiety categorical results with age and social anxiety as continuous variables. The results with age and social anxiety as continuous variables are now included in the manuscript and supplemental materials. To address the reviewer’s concern about the interpretation of the social anxiety data, we have revised the discussion section to match the results reported.

Say more in the introduction about why it might be that males tend to choose the harder task more often in the monetary EEfRT task.

We have now added extra information about why males tend to choose the harder task in the monetary EEfRT task in the introduction.

p. 3: To examine how potential social feedback influences the links individuals choose in social networks, we designed a paradigm analogous to the effort expenditure for rewards task (EEfRT). The EEfRT measures how much motoric effort humans are willing to expend to obtain monetary rewards (1, 2), and shows that participants reliably exert more effort for larger or more probable monetary rewards. Notably, males choose the higher effort task more often than females (1). One potential explanation may be that males have increased sensitivity to (monetary) rewards compared to females (3–5), but whether the EEfRT task is gender-biased is an open question (1).

The introduction should do more to contextualize potential interactive effects of age and social anxiety for peer-based social cognition. For example, see Smith, Nelson, Kircanski et al., 2020.

In the Introduction, we have added background literature on the development of social processing in adolescence and young adulthood, social feedback processing in individuals with social anxiety, and the potential interactive effects of age and social anxiety on peer-based social cognition. 

p. 4: Yet, social change in adolescence may coincide with the emergence of social anxiety (19–21). Individuals with social anxiety are averse to strangers and scrutiny and fear potential embarrassment or humiliation. Socially anxious individuals seem to fear social evaluation in general, both negatively and positively (22,23). Socially anxious youth also experience less positive affect after positive events or social interactions, and they may avoid potential positive interactions altogether to prevent disappointment or embarrassment (24,25). While adolescence is a phase of hypersensitivity to social evaluation in general, responsivity to social feedback seems to change as a function of age and severity of social anxiety (26,27). For instance, compared to older socially anxious youth (13-18 years), young socially anxious adolescents (8-13 years) were more responsive to unpredictable negative and predictable positive social evaluations, as indicated by more extreme pre-task ratings on peer likeability and greater activity in dorsal anterior cingulate cortex and insula during anticipation of social evaluation, suggesting increased salience of anticipated social interactions in young adolescents (28).

Were there sex differences on the Peterson Puberty Scale? Was puberty controlled for in analyses that consider sex?

There were no sex differences in pubertal development (t = 0.5, p = 0.618). Given that there were no sex differences in pubertal development, we did not control for puberty in the sex analyses. 

p.6, Table 1: There was no significant difference in pubertal development between males and females (p = 0.618).

Were there scoring differences in the adult and child version of the LSAS? If so, how were those differences handled?

There were no scoring differences between the child and adult version of the LSAS. Both versions yield scores with 24 rated items on a 0-3 Likert scale.

p. 6/7: Young adult participants completed the Liebowitz Social Anxiety Scale Self-Report (LSAS-SR; (32,33)) and adolescent participants completed the LSAS for Children and Adolescents Self-Report (LSAS-CA-SR; (34)) to assess social anxiety. The child and adult versions of the LSAS both yield scores with 24 rated items on a 0-3 Likert scale.

Please list the pre- and post-task questions verbatim. If a Likert scale was used, please describe the numeric and verbal anchors.

We now report the pre- and post-task questions in the methods section, including the 7-point Likert scale that was used. 

p. 7: Participants also completed three pre-task questions to determine the subjective value of receiving social media likes in general, the importance of the quantity of likes received, and the importance of the social status of the people liking the post. The three questions were: How much does it matter to you when someone likes your post on social media? When you post something on social media, how much does it matter to you how many likes you get? When someone likes your post on social media, how much does it matter to you how many followers that person has?. The three questions were rated on a 7-point Likert-scale (1- extremely unimportant to 7- extremely important).

p. 8: After completing the task, participants responded to three questions rated on a 7-point Likert scale that assessed the subjective importance of receiving a like during the task and how positive or negative participants felt when they received a like and when they did not. The three questions were: During the task, how much did it matter to you when someone liked your photo? How did you feel when someone liked your photo? How did you feel when someone did not like your photo?. 

Was the deception question simply a yes or no question? If not, what was the operational definition of deception?

Participants were asked whether they believed the task and asked to answer in either ‘yes’ or ‘no’. Participants were asked the following question: Did you think all the people liking your photos were real people?. Reasons of that participants did not believe the task were recorded (e.g., they thought none of the people were real or that it was some kind of computer program). 

p. 9: To assess whether participants believed the premise of the task, each participant was asked if they thought that each person viewing their photos during the game was a real person. Participants were asked the following question: Did you think all the people liking your photos were real people?

Please provide the logic for always assigning the easy task to the peer with a very low number of Instagram followers. Although this mirrors the classic EEfRT task, those unfamiliar with the task would benefit from further explanation of why it is set up this way.

The classic EEfRT task was indeed designed so that the easy task was paired with the lowest monetary reward. Our paradigm mirrors the EEfRT set-up, given that people may find positive social feedback from those with low numbers of followers less desirable than positive social feedback from those with many followers. As we are primarily interested in what factors (reward magnitude/ number of Instagram followers, or reward probability/ chance of positive feedback) lead people to choose the hard task (i.e. expend more effort to obtain positive social feedback), it is important that the easy task was not too appealing. 

p. 9: Analogous to the classic EEfRT, the easy task was always associated with an M-Turk user who had a very low number of Instagram followers (90 – 110), in order to make sure that the easy task was the least appealing compared to the hard task choices.

Are results consistent when age is used as a continuous rather than categorical measure? Please report on whether they are consistent and describe inconsistencies in the supplement.

Categorical age analyses were repeated with age as a continuous variable. Results are consistent, and are now in the supplemental material (Tables S3-S7) and referred to in the main manuscript.

p. 10/11: We conducted a 3x3 repeated measures ANOVA in R version 3.2.1 (R Core Team, 2015) with the proportion of hard task choices as the dependent variable and two within-subjects factors of social status level (low, medium, high) and probability level (12%, 50%, 88%), with age (adolescent, young adult) and sex (male, female) as between-subjects factors. Significant main effects and interactions (p < 0.05) were further interrogated using least-square means. The analysis was repeated with age as a continuous variable. 

p. 11: Secondary analyses including age as a continuous variable did not meaningfully change the results. The interaction between social status and probability (F (4, 688) = 2.7, p = 0.032) and between sex and social status remained (F (2, 688) = 11.3, p < .001). See S3 Table for full statistics. 

Likewise, are results consistent when the LSAS is treated as a continuous variable? Please indicate the N who fell above the clinical cutoff in each age group. No logic is provided for the use of two clinical cutoffs. Given there is a great deal of variability in anxiety symptoms in community samples, using the LSAS as a continuous variable would provide a more nuanced understanding of anxiety. It appears that one relation with LSAS as a continuous variable is provided in the supplement, but no mention of that occurs in the primary text.

First, we included the N that fell above each of the two cut-offs (60 and 47) in the manuscript and we now provide a rationale for using both clinical cut-offs:

p. 11: Second, for the social anxiety analyses, scores from the LSAS-SR and the LSAS-CA-SR were divided into two groups with a clinical cutoff of 60 (39,40). […] To confirm our findings, we performed two follow-up tests: First, we divided our sample with a different clinical cutoff of 47. A cut-off of 60 has been suggested to be the best cut-off for identifying generalized Social Anxiety Disorder (SAD), yet maximal sensitivity for generalized SAD was obtained using a cut-off of 47 (39,40). Given that this is a non-clinical sample of participants we chose to run all analyses using both cut-offs.

N’s for cut-off of 60, p. 11: Using this cutoff, 75 participants (32 adolescents (30 female) and 43 young adults (22 female)) fell into the non-elevated range (total scores ranging from 0-59) and 13 participants (8 adolescents (all female) and 5 young adults (3 female)) fell into the elevated range (total scores ranging from 60-103). 

N’s for cut-off of 47,p. 11: Using this cutoff, 59 participants (25 adolescents (24 female) and 34 young adults (17 female)) fell into the non-elevated range and 29 participants (15 adolescents (14 female) and 14 young adults (8 female)) fell into the elevated range. 

Categorical LSAS analyses were repeated with continuous LSAS scores. Results are consistent, and are now in the supplemental material and referred to in the main manuscript.

p. 11: Finally, social anxiety level was added to the analysis as a continuous variable. There were no meaningful differences between the results of the analyses using the clinical LSAS cutoff of 60, using a LSAS cutoff of 47, or including LSAS scores as a continuous variable (see S1-S7 tables). 

p. 11: Analyses were repeated at a different LSAS cutoff of 47 (a more inclusive definition of social anxiety) and with LSAS scores as a continuous variable, which yielded the same effects (see S6 and S7 Tables).

Secondary analyses that include interactive effects of sex, age, and anxiety should be described in the methods section. Given the small number of cases that fell above the clinical cutoff, these secondary analyses should be performed using age and anxiety as continuous variables.

The secondary analyses examine effects of sex, age, and LSAS scores have now been described more carefully in the methods section. As described in response to the reviewer’s previous two suggestions, additional analyses were performed with continuous age and social anxiety scores. These analyses with continuous age and LSAS scores are now mentioned in the methods and results and statistics of these analyses are reported in the S7 Supplemental Table.

p.10/11, methods: We conducted a 3x3 repeated measures ANOVA in R version 3.2.1 (R Core Team, 2015) with the proportion of hard task choices as the dependent variable and two within-subjects factors of social status level (low, medium, high) and probability level (12%, 50%, 88%), with age (adolescent, young adult) and sex (male, female) as between-subjects factors. Significant main effects and interactions (p < 0.05) were further interrogated using least-square means. The analysis was repeated with age as a continuous variable. 

p. 11, methods: Finally, social anxiety level was added to the analysis as a continuous variable. There were no meaningful differences between the results of the analyses using the clinical LSAS cutoff of 60, using a LSAS cutoff of 47, or including LSAS scores as a continuous variable (see S1-S7 tables).

p. 12, results: Secondary analyses including age as a continuous variable did not meaningfully change the results. The interaction between social status and probability (F (4, 688) = 2.7, p = 0.032) and between sex and social status remained (F (2, 688) = 11.3, p < .001). See S3 Table for full statistics. 

p. 13, results: Analyses were repeated at a different LSAS cutoff of 47 (a more inclusive definition of social anxiety) and with LSAS scores as a continuous variable, which yielded the same effects (see S6 and S7 Tables).

The significance indicators in Figure S3 do not always seem to map onto the lower order results described in the supplemental tables. For instance, social anxiety x probability data seem to be missing from the tables.

The reviewer correctly noted that the significance indicators were not completely in line with the statistics reported in the tables. First, all two-way interactions are now reported in the tables (S1-S7 Supplemental Tables). Further, the statistics are now presented in Figure 3 in the main manuscript. Supplemental Figure S3 has been removed as it was now redundant.

The authors characterize effects for social anxiety as follows: “Participants with higher symptoms of social anxiety avoided selecting high-status accounts and favored low-status accounts, relative to participants without symptoms of social anxiety, but differences only reached trend level in post-hoc comparisons (p’s > 0.084).” This does not fully capture the data described in the supplement – that there were no effects of social status among those with social anxiety. Only non-anxious individuals differentiated by social status. These results need to be better described in the text. The non-significant results currently highlighted in the text should be removed.

We agree with the reviewer and have changed the description of our findings to better reflect the post-hoc statistics. The non-significant findings have been removed from the text.

p. 13: Post-hoc comparisons showed that there were no significant group differences between those who did or did not have elevated social anxiety. However, within groups, adolescents and young adults without social anxiety symptoms chose the hard task more often with increasing social status, whereas those with elevated social anxiety symptoms did not differentiate by social status in their hard task choices.

In the discussion the authors highlight the idea that men (change to males – these data average across children and adults) are more likely to exert more effort for a social reward than women (change to females). However, in the introduction the authors indicate the same relation is observed in the monetary domain. Thus, it is impossible to make a claim that this is specific to social experiences without a direct comparison between tasks. A direct comparison across social and monetary tasks is a critical next step that should be discussed. This is mentioned in the discussion, but could be highlighted further.

The reviewer is indeed correct that no direct comparisons were made between effort to obtain rewards in the monetary or social domain. We have now further highlighted the need for future work directly comparing different types of rewards. We’ve also expanded our discussion referring to the literature on age and sex differences in the valuation of different types of rewards (e.g., social and/or monetary).

p.15: Second, males were substantially more likely than females to choose high-status associated with the hard task (50% vs 25% at the high-status level, respectively), though females still demonstrated a differential effect of status (comparing low-to-high). The sex difference is compatible with studies finding that men tend to be more competitive in physical effort-based tasks (73), yet rewards in these studies were monetary. While the tendency for competitiveness in males may have been a factor driving the increased effort to obtain social feedback, it has been shown that across age and between sexes there are behavioral and neurobiological differences in how social or financial (or other) rewards are valued relative to each other (7,51,74,75). Future work that directly compares effort-based decision-making for social versus financial or other rewards can determine whether and how adolescence and/or sex influence motivation to obtain specific types of rewards (37). 

In the discussion, it is important to note that it appears that effort is malleable among non-anxious individuals. Anxious individuals seem relatively unaffected by contextual factors (at least based on the data reported about status in the supplemental materials – probability data were not reported – perhaps this is where this effect may have emerged).

The full statistics, now also including the interaction between social anxiety and probability of positive feedback (p = 0.713), are reported in supplemental table S4. The findings suggest that while anxious individuals, as compared to non-anxious individuals, do not change the amount of effort in response to social status, they do still show the same pattern of increased effort for increased probability of positive social feedback. In terms of context, only the social status of the peer influences effort, whereas probability of positive social feedback in this case does not. We have changed the following section in the Discussion:

p. 15/16: It is also worth noting that there were no significant interactions between social anxiety and the probability of receiving positive social feedback or between social anxiety and sex in our data. The observation that in socially anxious individuals only the social status of the peer influenced effort, whereas probability of positive social feedback did not, may suggest that also in social anxiety effort is malleable, depending on context. Further, given the group sizes of individuals with elevated symptoms of social anxiety in the present study (conservative criterion size 13/89 subjects, 8 female, or liberal criterion size 29/89 subjects, 15 female), future work that explores the relationship between social anxiety and sex on social motivation will be important, as there is generally a higher preponderance of social anxiety in women. 

This line in the discussion is quite misleading: “Second, participants with high anxiety scores, compared to non-anxious subjects, were more likely to select low-status and to avoid high- status choices, causing them overall to display no modulation of choice by status (in contrast to non-anxious participants, who scaled effort to status).” As described by the author’s own supplemental results, there were no significant between group differences. This type of language needs to be eliminated from the discussion and abstract. Effects were primarily driven by status-related differences within the non-anxious group. Also – please change the language from “normal” to “non-elevated”.

We have changed “normal” to “non-elevated” throughout the manuscript and supplement. Further, also in line with the reviewer’s previous comment on the fact that the findings show that non-anxious individual differentiate by social status whereas socially anxious individuals don’t, we have changed the following sections in the abstract and discussion:

Abstract: Individuals with higher self-reported symptoms of social anxiety did not follow the typical pattern of increased effort to obtain social feedback from high status peers.

p. 15, Discussion: A result of particular interest is that participants with higher levels of self-reported social anxiety displayed no modulation of hard task choice by status, in contrast to non-anxious participants, who scaled effort to status.

 “The increase in motivation for low social status peers may be a safer or more secure social decision, and is consistent with prior work showing that individuals with higher anxiety symptoms are more risk- averse (33) and show fear-avoidant decision-making behaviors (34).” It is not clear what increase in motivation the authors are referring to – there is no difference within the high anxiety group for any pair-wise comparison of status level, nor is there a difference between the low and high anxiety groups at any status level.

In line with previous changes in the interpretation of this finding, this sentence now reads:

p. 15: The relative absence of differentiation in motivation, where socially anxious youth do not seem to follow the typical pattern of increased effort to obtain social feedback from high status peers, may be a safer or more secure social decision, and is consistent with prior work showing that individuals with higher anxiety symptoms are more risk-averse (76) and show fear-avoidant decision-making behaviors (77).

Reviewer #2: Thank you for the opportunity to review this manuscript entitled “An Effort-Based Social Feedback Paradigm Reveals Aversion to Popularity in Socially Anxious Participants and Increased Motivation in Adolescents” for consideration at PLOS ONE. As stated by the authors, this study created a novel social feedback program to assess motivational and effort-based ties to social links in adolescents and adults with and without high social anxiety symptoms.

This is an interesting topic, and in general I found this study to be a good contribution to the literature. Overall this manuscript is likely to be of interest to the broad audience at PLOS ONE, and has particular relevance for social media and technology-based approaches to future treatment. Having carefully considered this study, I provide (enumerated below) a few suggestions that may be of use to the authors in their efforts to further refine their manuscript. I provide one substantive comment regarding the introduction section below as well, that I highly suggest the authors consider:

We thank the reviewer for their positive comments on the relevance of our study and their helpful suggestions for improving the introduction and discussion. We have restructured our discussion, and provided extra background in the introduction, and more relevant literature to support our findings in the discussion.

1. Within the introduction, the authors identified potential differences in the literature that may contribute to adolescent versus adult social processing as well as some information regarding social anxiety and the impact that those symptoms may have on social processing. Although the authors present this information, it is extremely brief. I would strongly suggest that the authors significantly enhance their introduction section to set up a more solid rationale for the study. As it is currently written, they do not provide a compelling rationale that sets their study apart from others currently in circulation. Specifically, the authors should flesh out the theoretical underpinnings of adolescent versus adult social processing, as well as social feedback processing in individuals with social anxiety. Further, it would be beneficial for the authors to discuss other social feedback and decision-making paradigms that have been used in both adult and adolescent populations.

While there is indeed a large body of work investigating social feedback processing in adolescence and social anxiety, our study adds to the literature by introducing an ecologically valid paradigm that specifically captures social feedback in an online environment, which is relevant given the ever increasing amount of online social interactions. Also in response to Reviewer 1’s comments, we have now added background in the Introduction on the development of social processing in adolescence and young adulthood, social feedback processing in individuals with social anxiety, and as this reviewer suggested, the potential interactive effects of age and social anxiety on peer-based social cognition. We now also discuss other social feedback and decision making paradigms in the Discussion, which we have further highlighted under the next comment.

p. 3/4: We focused on behavioral effects in two populations of particular interest from a social feedback perspective: effects in adolescents relative to young adults, and effects in participants with social anxiety symptoms. Adolescence is a transformative period characterized by profound social development (6–8). During adolescence the importance of peer relationships strongly increases (9), yet with simultaneously growing self-consciousness (10,11) adolescents find themselves highly sensitive to peer approval and rejection (12–15). As such, adolescents may be more motivated to obtain social evaluative information (13,16). Previous work using physical exertion as a measure of motivation has shown that both adolescents and adults expend increasing effort to obtain increasing monetary rewards (17). However, given that adolescents are also more sensitive to online peer acceptance and rejection (10,18), adolescent users of social media might show greater motivation, i.e. willingness to expend effort, for positive social feedback, especially from those with high social status, than adults.

Yet, social change in adolescence may coincide with the emergence of social anxiety (19–21). Individuals with social anxiety are averse to strangers and scrutiny and fear potential embarrassment or humiliation. Socially anxious individuals seem to fear social evaluation in general, both negatively and positively (22,23). Socially anxious youth also experience less positive affect after positive events or social interactions, and they may avoid potential positive interactions altogether to prevent disappointment or embarrassment (24,25). While adolescence is a phase of hypersensitivity to social evaluation in general, responsivity to social feedback seems to change as a function of age and severity of social anxiety (26,27). For instance, compared to older socially anxious youth (13-18 years), young socially anxious adolescents (8-13 years) were more responsive to unpredictable negative and predictable positive social evaluations, as indicated by more extreme pre-task ratings on peer likeability and greater activity in dorsal anterior cingulate cortex and insula during anticipation of social evaluation, suggesting increased salience of anticipated social interactions in young adolescents (28). Users of social media with social anxiety may avoid linking to accounts with large numbers of followers, thereby limiting potential harm (and benefit), and they may be especially likely to seek ties likely to give positive feedback. Conversely, given that social milieus now increasingly assume an online form (29,30), social anxiety may be attenuated in the anonymity of online interactions (31). Whereas classically socially anxious individuals were described as people likely to avoid parties or public speaking, socially anxious individuals may show no differences in online social media interactions to those without anxiety. Because online social interactions are now an integral part of personal and professional life, it is important to understand what motivates or inhibits socially anxious youth and young adults so that mitigation strategies can be targeted (e.g., to prevent avoidable professional limitation).

2. Somewhat related to my first comment above, the discussion section of this manuscript could be expanded significantly to reflect the novelty of the paradigm the authors have created, and how it could compare to currently existing paradigms. I suggest the author’s consider this point.

In response to comments from both reviewers we have significantly restructured and expanded the discussion (p. 13-16). We agree with the reviewer that it is beneficial to our discussion to better compare our findings against previous work with different social feedback or decision making paradigms. We now also explicitly discuss why our social effort paradigm is a valuable addition as a tool to measure willingness to expend effort to obtain social feedback. Specifically:

p. 14: Overall, high social status increased the likelihood that participants would choose the hard task. These findings are consistent with financial incentive tasks in which participants give more effort for larger magnitude rewards (41,42), with research indicating that adolescents and young adults are drawn to photographs on Instagram by popularity (43,44), and that visual attention is greater for high- versus low-status peers (45–47). The probability of a positive reward also increased the decision to choose the hard task, consistent with earlier work using the original monetary version of the EEfRT (2,48–50). As such, our findings suggest that we developed a successful social adaptation of the original monetary EEfRT. Previous work using e.g. social incentive delay paradigms (51–54) or social feedback paradigms (28,55–58) have shown effects of adolescence or social anxiety on the processing of social evaluative information. Our social effort paradigm extends this work by capturing the motivation or willingness to expend effort to obtain social feedback. Yet importantly, how we value different types of social or non-social rewards, and consequently, how much effort we are willing to expend is influenced by individual characteristics such as age, gender, or social interaction- or anxiety problems (51,53,59–61). Below we will highlight several notable modulations of status pursuit in our data, all of which deserve further study. 

3. Within the discussion section, the authors jump to discussing neuroimaging studies using this paradigm. While I think this is more than appropriate, it does not directly link to the preceding sections. I suggest the authors thread this point into their discussion section more, and then elaborate on the novelty of this paradigm for neuroimaging studies.

We have restructured the discussion to better discuss the different factors (e.g. age, sex and social anxiety) that influence effort. As we agree with the reviewer that it is interesting and important for future studies to translate our findings to existing neuroimaging literature, we have now better integrated neurobiological findings in our discussion. We have changed the following sections in our Discussion:

p. 14/15: Only adolescents displayed an exception to a monotonic increase of effort with probability of positive feedback, displaying the highest effort when the probability of positive feedback was 50%. The 50% trials have the least assured outcome, and, are in that sense, the riskiest trials. As we found no self-reported differences in how adolescents value social media feedback compared to adults, it seems unlikely that the increased effort by adolescents was simply driven by a difference in social valuation, and we instead interpret the finding as potentially reflecting a difference in social motivation, consistent with findings from behavioral and neuroimaging studies. Interacting with peers is more salient and activating for adolescents (13,62) and adolescents are more willing to take risks in social contexts (63–65). Adolescence is a phase that is in general associated with heightened activity of reward and socioemotional processing circuitry (8), acceptance from peers activates dopamine rich frontostriatal circuitry (66), and adolescent risk-taking in social contexts has been associated with increased engagement of reward, limbic, and salience circuitry (67–70). The neural mechanism for the financial effort tasks, similar to our paradigm, relies on dopamine circuitry (41,71) and circuitry for valuation and salience (72), which all experience protracted development throughout adolescence into early adulthood. Second, males were substantially more likely than females to choose high-status associated with the hard task (50% vs 25% at the high-status level, respectively), though females still demonstrated a differential effect of status (comparing low-to-high). The sex difference is compatible with studies finding that men tend to be more competitive in physical effort-based tasks (73), yet rewards in these studies were monetary. While the tendency for competitiveness in males may have been a factor driving the increased effort to obtain social feedback, it has been shown that across age and between sexes there are behavioral and neurobiological differences in how social or financial (or other) rewards are valued relative to each other (7,51,74,75). Future work that directly compares effort-based decision-making for social versus financial or other rewards can determine whether and how adolescence and/or sex influence motivation to obtain specific types of rewards (37). 

A result of particular interest is that participants with higher levels of self-reported social anxiety displayed no modulation of hard task choice by status, in contrast to non-anxious participants, who scaled effort to status. This finding held using both conservative and liberal criteria to define anxious and non-anxious groups. The relative absence of differentiation in motivation, where socially anxious youth do not seem to follow the typical pattern of increased effort to obtain social feedback from high status peers, may be a safer or more secure social decision, and is consistent with prior work showing that individuals with higher anxiety symptoms are more risk-averse (76) and show fear-avoidant decision-making behaviors (77). Neuroimaging studies have shown changes in striatal activity during positive and negative social feedback in socially anxious adolescents, which has been suggested to reflect increased affective arousal during social evaluation (28,55). Our findings may also reflect less context-appropriate social decision-making and an increase in rigidity in social scenarios, rather than flexibly adapting to varying feedback that one might receive during social interactions. Their altered decision-making pattern may also be maladaptive and perpetuate symptoms of loneliness and isolation, and thus may be an important target for interventions.

---

## [Decision Letter · Decision Letter 1]

16 Mar 2021

An effort-based social feedback paradigm reveals aversion to popularity in socially anxious participants and increased motivation in adolescents

PONE-D-20-26279R1

Dear Dr. Bos,

We’re pleased to inform you that your manuscript has been judged scientifically suitable for publication and will be formally accepted for publication once it meets all outstanding technical requirements.

Kind regards,

David V. Smith, Ph.D.

Academic Editor

PLOS ONE

Additional Editor Comments (optional):

Reviewers' comments:

Reviewer's Responses to Questions

**Comments to the Author**

1. If the authors have adequately addressed your comments raised in a previous round of review and you feel that this manuscript is now acceptable for publication, you may indicate that here to bypass the “Comments to the Author” section, enter your conflict of interest statement in the “Confidential to Editor” section, and submit your "Accept" recommendation.

Reviewer #1: All comments have been addressed

Reviewer #2: All comments have been addressed

2. Is the manuscript technically sound, and do the data support the conclusions?

Reviewer #1: Yes

Reviewer #2: Yes

3. Has the statistical analysis been performed appropriately and rigorously? 

Reviewer #1: Yes

Reviewer #2: Yes

4. Have the authors made all data underlying the findings in their manuscript fully available?

Reviewer #1: Yes

Reviewer #2: Yes

5. Is the manuscript presented in an intelligible fashion and written in standard English?

Reviewer #1: Yes

Reviewer #2: Yes

6. Review Comments to the Author

Reviewer #1: The authors did an excellent job addressing all of my concerns. This paper has the potential to make an important contribution to the field. Great work!

Reviewer #2: (No Response)

7. PLOS authors have the option to publish the peer review history of their article (what does this mean?). If published, this will include your full peer review and any attached files.

Reviewer #1: No

Reviewer #2: No

---

## [Editor Report · Acceptance letter]

16 Apr 2021

PONE-D-20-26279R1 

An effort-based social feedback paradigm reveals aversion to popularity in socially anxious participants and increased motivation in adolescents 

Dear Dr. Bos:

I'm pleased to inform you that your manuscript has been deemed suitable for publication in PLOS ONE. Congratulations! Your manuscript is now with our production department. 

Kind regards, 

on behalf of

Dr. David V. Smith 

Academic Editor

PLOS ONE